# Evaluating the Effects of Kidney Preservation at 10 °C with Hemopure and Sodium Thiosulfate in a Rat Model of Syngeneic Orthotopic Kidney Transplantation

**DOI:** 10.3390/ijms25042210

**Published:** 2024-02-12

**Authors:** Maria Abou Taka, George J. Dugbartey, Mahms Richard-Mohamed, Patrick McLeod, Jifu Jiang, Sally Major, Jacqueline Arp, Caroline O’Neil, Winnie Liu, Manal Gabril, Madeleine Moussa, Patrick Luke, Alp Sener

**Affiliations:** 1Department of Microbiology and Immunology, Schulich School of Medicine and Dentistry, Western University, London, ON N6A 5C1, Canada; mabouta2@uwo.ca; 2Matthew Mailing Centre for Translational Transplant Studies, London Health Sciences Centre, London, ON N6A 5A5, Canada; gdugbart@uwo.ca (G.J.D.); mahmoud.richard-mohamed@lhsc.on.ca (M.R.-M.); patrick.luke@lhsc.on.ca (P.L.); 3Multi-Organ Transplant Program, London Health Sciences Centre, London, ON N6A 5A5, Canada; 4Department of Pharmacology and Toxicology, School of Pharmacy, College of Health Sciences, University of Ghana, Legon, Accra P.O. Box LG 1181, Ghana; 5London Health Sciences Centre, Department of Surgery, Division of Urology, London, ON N6A 5A5, Canada; 6The Molecular Pathology Core, Robarts Research Institute, London, ON N6A 5A5, Canada; 7London Health Sciences Centre, Department of Pathology and Laboratory Medicine, London, ON N6A 5A5, Canadamanal.gabril@lhsc.on.ca (M.G.); madeleine.moussa@lhsc.on.ca (M.M.)

**Keywords:** sodium thiosulfate (STS), Hemopure, ischemia–reperfusion injury (IRI), static cold storage (SCS), kidney transplantation, graft and recipient survival

## Abstract

Kidney transplantation is preferred for end-stage renal disease. The current gold standard for kidney preservation is static cold storage (SCS) at 4 °C. However, SCS contributes to renal graft damage through ischemia–reperfusion injury (IRI). We previously reported renal graft protection after SCS with a hydrogen sulfide donor, sodium thiosulfate (STS), at 4 °C. Therefore, this study aims to investigate whether SCS at 10 °C with STS and Hemopure (blood substitute), will provide similar protection. Using in vitro model of IRI, we subjected rat renal proximal tubular epithelial cells to hypoxia–reoxygenation for 24 h at 10 °C with or without STS and measured cell viability. In vivo, we preserved 36 donor kidneys of Lewis rats for 24 h in a preservation solution at 10 °C supplemented with STS, Hemopure, or both followed by transplantation. Tissue damage and recipient graft function parameters, including serum creatinine, blood urea nitrogen, urine osmolality, and glomerular filtration rate (GFR), were evaluated. STS-treated proximal tubular epithelial cells exhibited enhanced viability at 10 °C compared with untreated control cells (*p* < 0.05). Also, STS and Hemopure improved renal graft function compared with control grafts (*p* < 0.05) in the early time period after the transplant, but long-term function did not reach significance. Overall, renal graft preservation at 10 °C with STS and Hemopure supplementation has the potential to enhance graft function and reduce kidney damage, suggesting a novel approach to reducing IRI and post-transplant complications.

## 1. Introduction

It is estimated that more than 2 million North Americans will be living with end-stage renal disease (ESRD) by 2030 [1]. Approximately 5 million people with ESRD require renal replacement therapy in the form of dialysis or kidney transplantation, thus illustrating the significant burden ESRD has on the global healthcare system [2,3,4,5]. Although dialysis is particularly effective for removing uremic toxins, achieving electrolyte and pH balance, and resolving fluid abnormalities [6], kidney transplantation is the preferred treatment for patients with ESRD [7,8].

Prior to kidney transplantation, the donor kidney is preserved in hypothermic solution to inhibit cellular processes and reduce metabolism [9,10]. This is advantageous because hypothermia slows biological processes (e.g., depletion of adenosine triphosphate) in organs that have been removed from their normal physiological environment and reduces ischemic injury. In fact, it has been shown that hypothermic preservation reduces the metabolic activity of donor organs by 2- to 3-fold for every 10 °C drop in temperature [11]. Logistically, hypothermic preservation allows time for staff and facility mobilization, organ transport, laboratory tests, and recipient selection and preparation. The current clinical standard of care for donor organ preservation is 4 °C on ice. However, emerging evidence shows that several metabolic pathways are affected during prolonged renal graft preservation at 4 °C, thereby contributing significantly to ischemia–reperfusion injury (IRI) and associated post-transplant complications [12,13,14,15,16]. For example, a study by Ojo and colleagues [14] showed that renal graft preservation at 4 °C increases the risk of delayed graft function (DGF) by 23% for every 6 h increase in cold ischemic time, while Barba et al. [15] also reported a 10% increase in DGF following each hour of cold ischemic time in deceased donor kidneys after 18 h of preservation at 4 °C. Hence, there is a critical need to modify the current preservation protocol to reduce IRI associated with 4 °C kidney storage.

A few preclinical attempts have been made to improve renal graft preservation at temperatures above 4 °C, such as 20 °C, 32 °C, and 35 °C, and have shown promising results [17,18,19]. Recent studies also showed that the preservation of human lungs, liver, and cardiac grafts at 10 °C improved organ function and enhanced patient survival [20,21,22]. However, this has yet to be tested in kidney transplantation. Therefore, the aim of our study was to evaluate the effects of renal graft preservation at 10 °C with a hydrogen sulfide (H_2_S) donor, sodium thiosulfate (STS), which was shown to reduce kidney tissue injury, and a bovine-derived blood substitute, Hemopure, which effectively delivers oxygen to ischemic tissues [23,24,25,26,27,28,29,30].

## 2. Results

### 2.1. 10 °C STS Treatment Improved Proximal Tubular Epithelial Cell Survival

As illustrated in Figure 1 and Figure 2, flow cytometry analysis of apoptosis and necrosis revealed that STS-treated cells at 10 °C had a significant increase in the proportion of viable cells (*p* < 0.05) compared with those treated at the other temperatures investigated (Figure 3B and Figure 4). The reverse trend was seen when observing the proportion of early (Figure 3C) and late (Figure 3D) apoptotic cells, where STS treatment at 10 °C exhibited a significant decrease in the percentage of apoptotic cells (*p* < 0.05) compared with the other temperatures studied. However, STS treatment at 10 °C significantly reduced the proportion of necrotic cells (*p* < 0.05) only compared with STS-treated cells at 37 °C (Figure 3E). 

### 2.2. Hemopure and STS-Supplemented Preservation Solution Improved Early Renal Graft Function

There was no significant difference between the different renal preservation treatments in recipient survival following kidney transplantation (Figure 5). As a result, we focused on measuring renal parameters soon after transplant surgery (i.e., POD 3) to determine potential differences in early graft outcomes. Blood and urine samples were collected on POD 3 after rats spent 24 h in the metabolic stage. Recipient rats whose renal grafts were preserved in Hemopure with 150 µM STS in UW solution exhibited markedly improved graft function during the early post-transplant period (POD 3) compared with rats receiving the other treatments, except for the Sham-operated rats. On POD 3, rats whose renal grafts were preserved with Hemopure and STS had significantly increased urine osmolality (*p* < 0.05) compared with rats undergoing the other treatments (Figure 6). Also on POD 3, animals treated with a combination of Hemopure and STS had significantly decreased serum levels of creatinine (*p* < 0.05) compared with rats undergoing the other preservation treatments (Figure 7A). Similarly, on POD 3, rats treated with the combination of Hemopure and STS had significantly decreased blood urea nitrogen (*p* < 0.05) compared with rats undergoing the other preservation treatments (Figure 7B). Further, the eGFR of the rats treated with a combination of Hemopure and STS was significantly improved on POD 3 (*p* < 0.05) compared with the rats receiving other treatments in the study (Figure 7C). 

### 2.3. Hemopure and STS-Supplemented Preservation Solution Reduced Donor Kidney Apoptosis and Necrosis after Kidney Transplantation

Kidney sections were obtained on POD 3 and stained with H&E, a measure of ATN, and were scored by two blinded renal pathologists. Rats whose renal grafts were preserved with Hemopure and 150 µM STS had significantly decreased ATN scores (*p* < 0.05) compared with those receiving UW- and Hemopure-with-UW-treated renal grafts, whose ATN scores were significantly higher (Figure 8). Additionally, kidney sections obtained on POD 3 were stained with TUNEL, a measure of apoptosis. Renal grafts preserved in Hemopure and STS had a significantly lower percentage of TUNEL-positive areas (*p* < 0.05) in the cortical and medullary areas of the kidney compared with grafts that underwent the other treatments, except for those of the Sham-operated rats, in the study (Figure 9). 

### 2.4. Hemopure and STS-Supplemented Preservation Solution Decreased Donor Kidney Injury Markers and Inflammatory Infiltrate after Kidney Transplantation 

To determine the impact of preserving renal grafts at 10 °C with UW preservation solution supplemented with Hemopure and 150 µM STS on renal graft injury, kidney sections obtained on POD 3 were stained with kidney injury molecule-1 (KIM-1) to detect proximal tubular injury. Compared with rats receiving the other treatments in the study, except for the Sham-operated rats, recipient rats whose renal grafts were preserved in Hemopure and 150 µM STS exhibited significantly reduced Kim-1-positive areas (*p* < 0.05) on POD 3 (Figure 10).

Additionally, renal grafts obtained on POD 3 were immunohistochemically stained with antibodies against the macrophage marker CD68 and the neutrophil marker myeloperoxidase (MPO). Compared with the grafts that underwent the other treatments investigated, except for the Sham group, renal grafts preserved in Hemopure and 150 µM STS exhibited significantly decreased (*p* < 0.05) percentages of both CD68- (Figure 11) and MPO-positive (Figure 12) areas in the kidney tissue.

## 3. Discussion

The findings of our in vitro model of rat renal IRI reveal that 150 µM STS treatment at 10 °C protected rat proximal tubular epithelial cells from IRI compared with the other temperatures investigated, including 4 °C, the current gold standard for renal graft preservation, 21 °C, the subnormothermic temperature our group previously studied, and 37 °C, the standard for normothermic preservation [31,32,33,34,35,36,37]. It was recently shown that, at 10 °C, essential molecular processes that maintain cell integrity and viability are active [38,39,40,41,42]. For instance, the urine-concentrating function of the kidney is effective at 10 °C with Hemopure and STS.In a study involving the genetic knockout of AQP-1 in mice, it was found that the ability of the kidney to concentrate urine was abolished [43]. In a mouse model of kidney injury, a recent study found that administration of a H_2_S donor, GYY4137, upregulated the protein expression of AQP-2, which is expressed in the collecting duct of the nephron, which, in turn, promoted urine concentration [44]. This could provide a possible explanation for the protective effect of STS potentially observed in this study. 

Also, preserving renal grafts with Hemopure and STS resulted in a significant decrease in serum levels of creatinine and BUN compared with the other treatments in the study. These findings are consistent with several studies that found that administration of H_2_S donors, such as NaHS, in rodent kidney injury models increased renal clearance and, thus, protected kidneys from high levels of serum creatinine and BUN [45,46,47]. Additionally, we found that rats whose renal grafts were preserved in Hemopure and STS had improved estimated GFR (eGFR). eGFR estimates the filtration rate of fluid in the kidney, which is widely used as a clinical assessment for kidney function. In a rat model of hypertension-induced kidney injury, Fan et al. [48] orally administered STS and found that STS-treated rats had significantly improved eGFR compared with the non-STS-treated rats. 

Renal graft preservation with Hemopure and STS resulted in significant protection from early tissue injury and inflammation compared with the other treatments in the study. These renal grafts had significantly decreased ATN scores (determined by H&E staining) and significantly decreased TUNEL-positive areas in the kidneys on POD 3. Chou et al. [49] similarly found that in a rat model of renal injury, STS administration reduced kidney tissue apoptosis and ATN. Another group at our center investigated the use of oxygenated Hemopure in an ex vivo porcine model of renal IRI at 22 °C and found that Hemopure-perfused kidneys were also associated with vastly reduced ATN scores compared with the SCS kidneys [50]. Rats whose renal grafts were preserved in Hemopure and STS also exhibited significantly decreased KIM-1-positive areas in the kidneys on POD 3. KIM-1 is a type 1 transmembrane protein that is upregulated in the proximal tubule [51,52]. In a rat ischemia model, Western blot analysis revealed that KIM-1 protein expression was robustly upregulated in injured proximal tubular epithelial cells, which is the region of the nephron most highly impacted by ischemic injury [53]. These findings are consistent with a previous study carried out by our group, where renal graft preservation with STS also resulted in significantly reduced KIM-1 expression in a rat model of kidney transplantation [54]. 

Finally, renal graft preservation in Hemopure and STS resulted in significantly downregulated CD68 and MPO expression on POD 3. CD68 is a marker for macrophages, whereas MPO is a marker for neutrophils. Macrophages and neutrophils are key components of the innate immune system that promote inflammation during renal IRI. Inflammation is a key initiator of the early events that leads to the multitude of intracellular responses to IRI, ultimately resulting in apoptosis and necrosis [55]. Similarly, Dursun et al. [46] looked at the effects of STS on a hypertension-induced kidney injury model in rats and found that STS treatment significantly reduced macrophage and neutrophil infiltration.

### Limitations

Despite the clinical relevance of our in vivo model of rat renal transplantation, the model was based on syngeneic transplantation to eliminate any confounding effects of immunosuppression. However, syngeneic transplantation is not clinically relevant in kidney transplant human patients, and perhaps an allogeneic model would have been more clinically applicable for this study. Next, there is a lack of quantitative approaches to measuring the concentrations of relevant molecules that could help us further understand the beneficial effects of STS and Hemopure from a mechanistic point of view. For instance, we did not measure plasma H_2_S concentration. Although there are several different methods of measuring the systemic concentration of H_2_S, such as high-performance liquid chromatography, mass spectrometry, and refractometry, the literature remains inconclusive regarding the most reliable method of accurately quantifying H_2_S levels, especially when released in small concentrations from slow-releasing H_2_S donors, like STS [56,57,58]. Additionally, we did not quantify renal blood flow and local differences in pO_2_ to determine changes in renal tissue oxygenation and consumption. This information would have been valuable in determining potential changes in kidney metabolism in response to perfusion and the subsequent storage of Hemopure-treated renal grafts at 10 °C. 

Further, while our results revealed beneficial early (i.e., POD 3) post-transplant outcomes following 10 °C storage in Hemopure and STS, the overall survival of the rats was poor. This may be in part due to the 24 h renal graft static storage time, which may have been too long, resulting in kidney damage that adversely impacted long-term survival in this model [59]. Additionally, machine perfusion may have been more efficient in supplying STS and oxygen to the renal graft to improve post-transplant survival. However, this technology was not available to us for a murine study model. As such, we will be conducting future studies in larger animals, such as porcine models, where machine perfusion is more feasible. Also, the plasma half-life of STS in murine models is only approximately 26 min, and perhaps post-transplant administration of STS would have been necessary to prolong the beneficial effects of STS in the recipient to improve survival [60].

## 4. Methods

### 4.1. In Vitro Model of Rat Renal IRI

A previously established in vitro model of rat renal IRI by our lab [23] was used to investigate the feasibility and protective effects of STS-supplemented solution during hypoxic storage at 4 °C, 10 °C, 21 °C, and 37 °C (Figure 1). NRK-52E cells (ATCC, Manassas, VA, USA), proximal tubular epithelial cells from rat kidneys, were used in this in vitro study since proximal tubule cells are especially susceptible to ischemic injury in IRI [25,26,27,28,29,30]. Experimental cells were treated with either fetal bovine serum (FBS)-free Dulbecco’s modified eagle medium (DMEM) media (SF) or SF with 150 µM STS. (Seacalphyx^®^ (Seaford Pharmaceuticals Inc., Mississauga, ON, Canada)). This dose of STS was used because our lab has previously shown that 150 µM STS has cytoprotective effects against the same cell line in a similar in vitro model of rat renal IRI [23]. These cells were incubated for 24 h at 4 °C, 10 °C, 21 °C, or 37 °C in hypoxic conditions (5% CO_2_, 0.5% O_2_, and 95% N_2_) to mimic ischemia during cold, subnormothermic, and normothermic renal preservation conditions. These experimental hypoxic conditions were established using the HypOxystation^®^ H85 hypoxia chamber (HypOxygen, Frederick, MD, USA). After hypoxic conditions, the media of the experimental cells were replaced with control media (DMEM with FBS), and the cells were reoxygenated via incubation for 24 h in normoxic conditions (37 °C, 21% O_2_, and 5% CO_2_) to mimic reperfusion in renal IRI.

### 4.2. Analysis of Cell Viability, Apoptosis, and Necrosis

The cells were analyzed for apoptosis and necrosis after reoxygenation in normoxic conditions using FITC-conjugated Annexin-V (FITC–Annexin-V) (BioLegend, San Diego, CA, USA) and PerCP-conjugated Propidium Iodide (PerCP-PI) (BioLegend, USA), respectively. To optimize detection, all samples were incubated at room temperature in the dark for 15 min. After, the stained samples were analyzed for cell viability, apoptosis, and necrosis using the CytoFLEX S (Beckman Coulter, Brea, CA, USA). One sample of unstained live cells and two samples of heat-killed cells (90 °C, 12 min), stained with either FITC–Annexin-V or PerCP-PI, were used for compensation. FlowJo^®^ V11 (FlowJo LLC, Ashland, OR, USA) was used to appropriately gate the data for statistical analysis. 

### 4.3. Experimental Animals

A previously established in vivo syngeneic model of rat renal transplantation was used for this study [23]. Male Lewis rats were purchased from Charles River Canada (St. Constant, QC, Canada) and used at 250–300 g (*n* = 36). Rats were maintained in the Animal Care and Veterinary Services (ACVS) facility at Western University (London, ON, Canada) under standard conditions. The animal studies were approved by the Western University Council on Animal Care and Animal Use (Protocol ID: 2018-155). 

### 4.4. Syngeneic Orthotopic Kidney Transplant Model

Syngeneic Lewis rat renal transplantation was performed to eliminate any potential confounding effects of immunosuppression. Rats were randomized into treatment groups, anesthetized with ketamine (30 mg/kg), and maintained under anesthesia with isoflurane during kidney transplant surgery. Using aseptic techniques, the left donor kidneys were procured and flushed with a 28 G Angiocath Becton-Dickinson with 5 mL of one of the following treatments until the venous effluent was clear: UW preservation solution (*n* = 5), UW plus 150 µM STS (*n* = 4), UW plus Hemopure (*n* = 4), or UW plus Hemopure and 150 µM STS (*n* = 5). Kidney grafts were then stored in 50 mL of the same perfusion solution in a sterile 50 mL conical tube at 10 °C for 24 h; this is the length of time that was previously shown to result in acute tubular necrosis and inflammation that led to graft function loss [58]. After undergoing bilateral nephrectomy, recipient rats underwent renal transplantation with donor kidneys removed from 10 °C storage by end-to-side anastomosis of donor inferior vena cava to recipient inferior vena cava with an 11-0 Prolene suture followed by the donor ureter being anastomosed to the recipient ureter using 10-0 PDS sutures. Since our aim was to assess the protective effects of STS and Hemopure on renal graft function after extreme 10 °C storage, we used a survival model where the recipient rat was only dependent upon the transplanted kidney for renal function and survival. Sham-operated rats (midline incision only; *n* = 5) were used to establish a baseline for survival, histological analysis, blood urea nitrogen (BUN), and serum creatinine. All surgeries were performed by the same microsurgeon, who was blinded to the experimental design, with the length of surgery for the recipient being 3 h for all treatment groups. There was no difference in operating times between all the treatment groups. Graft failure was presumed in animals that required premature sacrifice (severe respiratory distress) or death. 

### 4.5. Creatinine and Urine Assays

After renal transplantation, recipient rats were monitored in metabolic cages for 14 days or until sacrifice. To examine early graft outcomes, on POD 3, recipient rats were removed from their metabolic cage and the previous 24 h’ worth of urine output was collected for urine osmolality analysis. Also on POD 3, 150 µL of blood was taken from the tail vein of the recipient rat following the ACVS protocol. The blood samples were centrifuged at 2000× *g* for 10 min at 4 °C to separate the serum to determine the levels of serum creatinine and BUN using the enzymatic method on the IDEXX Catalyst One Chemistry Analyzer machine (IDEXX, Markham, ON, Canada). Urine osmolality levels were determined by freezing-point osmometry using the 3320 Osmometer machine (Advanced Instruments, Norwood, MA, USA) and compared with company-provided standards. On POD 3, the estimated glomerular filtration rate (eGFR) was calculated using serum creatinine (*C*), BUN (*U*), and the mass (*W*) of the rats using the following equations [59]:(1)$$Serum\;Creatinine<52\frac{\mu mol}{L}:eGFR=880\times W^{0.695}\times C^{-0.660}\times U^{-0.391}$$
(2)$$Serum\;Creatinine\geq 52\frac{\mu mol}{L}:GFR=5862\times W^{0.695}\times C^{-1.150}\times U^{-0.391}$$
where Equation (1) accounts for measured POD 3 serum creatinine levels less than 52 µmol/L and Equation (2) accounts for measured POD 3 serum creatinine levels greater than or equal to 52 µmol/L. 

### 4.6. Histopathological Imaging and Quantification

For each treatment in this study, renal grafts obtained at the time of sacrifice on POD 3 were placed in 10% formalin for a minimum of 72 h for paraffin embedding and sectioning. Formalin-fixed kidney sections, including cortex and medulla, were stained with Hematoxylin and Eosin (H&E) and Terminal deoxynucleotidyl transferase dUTP nick end labeling (TUNEL) to determine levels of acute tubular necrosis (ATN) and apoptosis, respectively. To quantify ATN, H&E slides were scored by two blinded renal pathologists as per the following scheme: 1 = <11%, 2 = 11–24%, 3 = 25–45%, 4 = 46–75%, 5 = >75% graft ATN [60].

Histological sections also underwent immunohistochemical staining, where sections were incubated with antibodies against kidney injury marker-1 (KIM-1), macrophage surface marker (CD68), and the neutrophil-specific enzyme myeloperoxidase (MPO) and visualized with secondary antibodies and DAB (3,3′-Diaminobenzidine) substrate chromogen using the Dako Envision System (Dako, Glostrup, Denmark) as per the manufacturer’s protocol. A Leica SP8 Confocal Microscope (Leica Camera AG^®^, Wetzlar, Germany) was used to randomly capture ten images per section. All images were captured at 20× magnification. The images were run through Image J v.1.53 (National Institutes of Health, Bethesda, MD, USA) to determine the total proportion (%) of TUNEL+, KIM-1+, CD68+, and MPO+ areas for each kidney section.

### 4.7. Statistical Analysis

GraphPad Prism 9 was used to create graphs and conduct statistical analyses for this study. Survival data were assessed using Kaplan–Meier survival analysis. All other data were analyzed using one-way ANOVA followed by Tukey’s post hoc test for comparisons of three or more experimental groups. Statistical significance was accepted at *p* < 0.05.

## 5. Conclusions

Overall, our results reveal that preserving renal grafts at 10 °C with Hemopure and STS enhanced key early kidney graft parameters, as observed by reduced renal tissue damage and enhanced kidney functional outcomes on POD 3. However, longer-term graft function did not show statistical significance. Importantly, we developed a novel oxygenation circuit to allow us to be the first documented group to evaluate renal graft preservation at 10 °C. This novel circuit could shift the paradigm away from the costly setup of machine perfusion to a device that is more cost-effective and more feasible to use, especially at subnormothermic temperatures such as 10 °C, and widely accessible for nations around the globe, along with our modified fluid, which could be used in perfusion pumps, with the possibility of storing the graft at temperatures higher than the widely used temperature of 4 °C without losing its function after reperfusion. However, it is important to note that the main premise of this work was to find an equivalent to the traditional static cold storage of renal grafts at 4 °C by designing a novel platform of kidney oxygenation at 10 °C. As such, further innovative studies are required to facilitate the clinical implementation of 10 °C renal graft preservation with Hemopure and STS in practice. Nevertheless, the experimental findings of this study will inform future studies that are seeking to advance kidney preservation techniques with H_2_S therapeutics. With the growing incidence of DGF associated with cold IRI, and the ever-increasing number of patients with ESRD who require kidney transplantation, it is critical that future studies continue to improve kidney preservation techniques to enhance the quality of life and longevity of kidney transplant recipients.

## Figures and Tables

**Figure 1 ijms-25-02210-f001:**
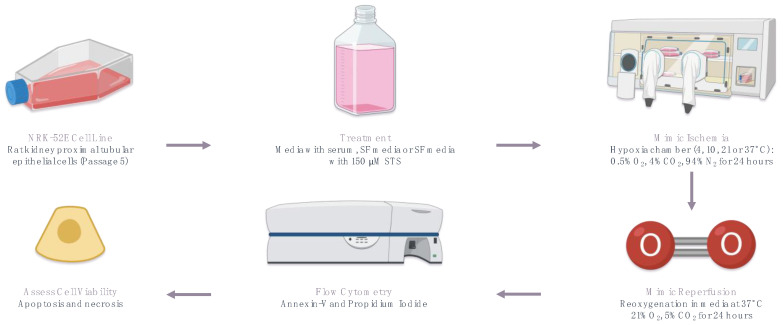
In vitro model of rat renal IRI. A summary of the in vitro rat renal IRI model used to determine the protective effects of STS at various temperatures (4 °C, 10 °C, 21 °C, and 37 °C). Figure prepared with BioRender (biorender.com). Accessed on 1 March 2023.

**Figure 2 ijms-25-02210-f002:**
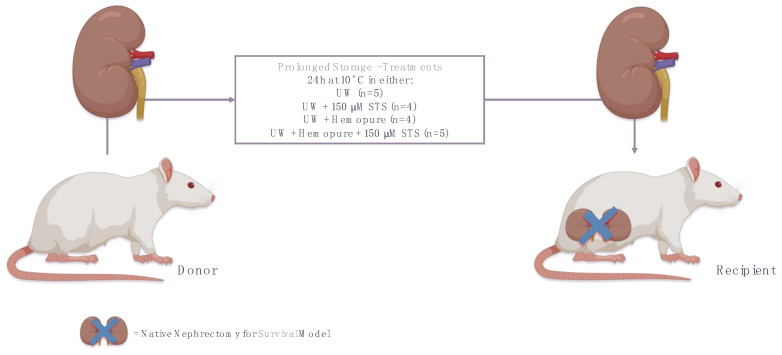
In vivo syngeneic model of renal IRI in a rat kidney transplantation model at 10 °C. A summary of the experimental procedure used to determine renal graft outcomes following prolonged kidney storage for 24 h at 10 °C with UW, UW with 150 µM STS, Hemopure with UW, and Hemopure with UW and 150 µM STS. Figure prepared with BioRender (biorender.com). Accessed on 1 March 2023.

**Figure 3 ijms-25-02210-f003:**
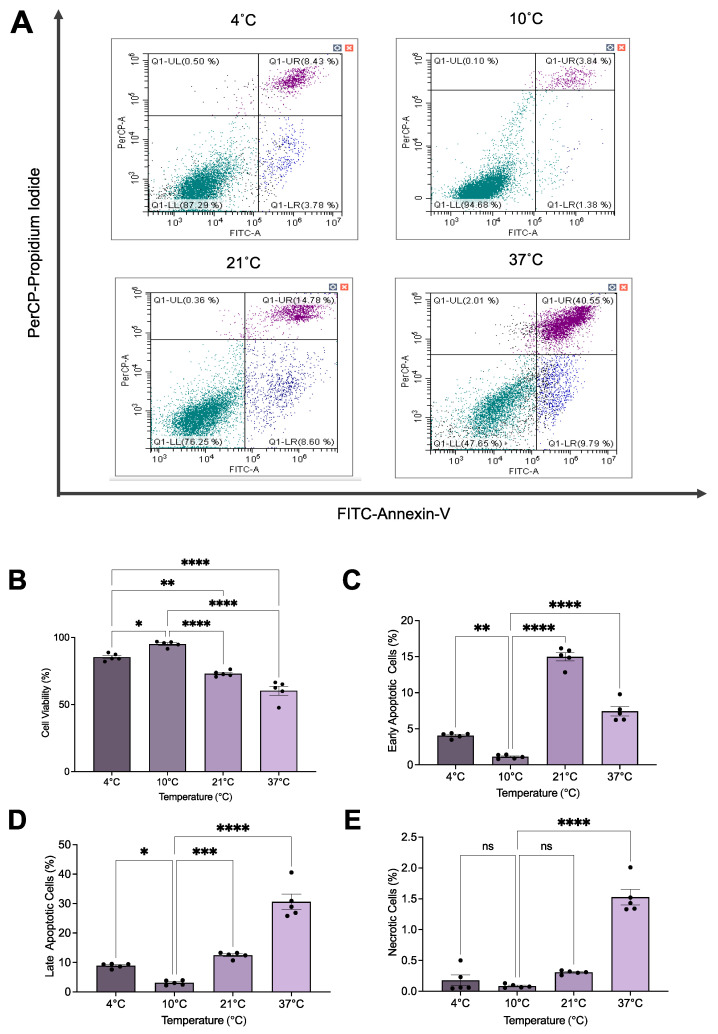
The 10 °C STS treatment improved rat proximal tubule epithelial cell survival during renal IRI. (**A**) Representative flow cytometry images for all temperatures investigated. (**B**) Mean cell viability (%) (*n* = 5) as determined by the proportion of cells negative for FITC–Annexin-V and PerCP-PI staining. (**C**) Mean early apoptosis (%) (*n* = 5) as determined by the percentage of cells positively stained for FITC–Annexin-V and negatively stained for PerCP-PI. (**D**) Mean late apoptosis (%) (*n* = 5) as determined by the percentage of cells positively stained for FITC–Annexin-V and PerCP-PI. (**E**) Mean necrosis (%) (*n* = 5) as determined by the percentage of cells negatively stained for FITC–Annexin-V and positively stained for PerCP-PI. Lines indicate mean ± SEM. Means were analyzed using one-way ANOVA and Tukey’s post hoc test. *, *p* < 0.05. **, *p* < 0.01. ***, *p* < 0.001. ****, *p* < 0.0001. ns, not significant (*p* > 0.05).

**Figure 4 ijms-25-02210-f004:**
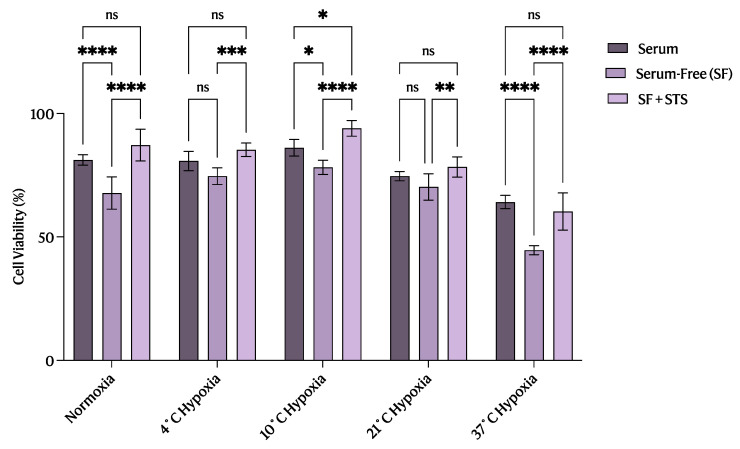
The 10 °C STS treatment improved rat proximal tubule epithelial cell viability during renal IRI compared with control cells. Mean cell viability (%) for serum-treated cells (positive control; *n* = 5), serum-free (SF) treated cells (negative control; *n* = 5), and STS-treated cells (*n* = 5) as determined by the proportion of cells negative for FITC–Annexin-V and PerCP-PI staining. Lines indicate mean ± SEM. Means were analyzed using one-way ANOVA and Tukey’s post hoc test. *, *p* < 0.05. **, *p* < 0.01. ***, *p* < 0.001. ****, *p* < 0.0001. ns, not significant (*p* > 0.05).

**Figure 5 ijms-25-02210-f005:**
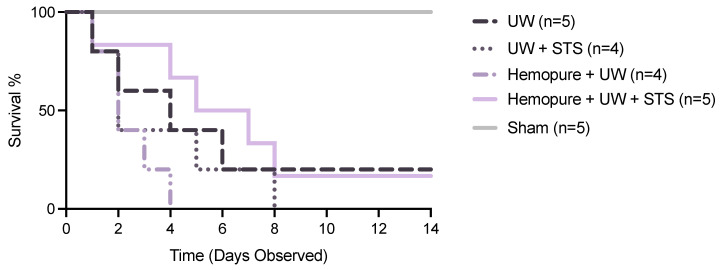
Renal graft preservation at 10 °C modulated recipient survival following transplantation. Survival rates of kidney transplant recipients whose donor kidneys were perfused and stored for 24 h at 10 °C in UW (*n* = 5), UW + 150 µM STS (*n* = 4), Hemopure + UW (*n* = 4), or Hemopure + UW + 150 µM STS (*n* = 5) as well as those of Sham-operated rats. Survival data analyzed via Kaplan–Meier survival analysis and log-rank test. Note: one UW rat and one Hemopure + UW + STS rat survived until POD 14.

**Figure 6 ijms-25-02210-f006:**
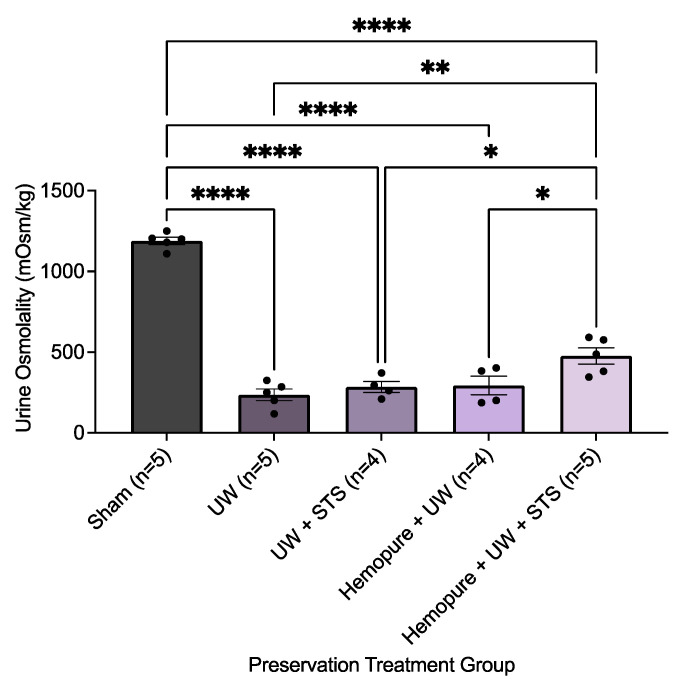
Renal graft preservation at 10 °C with Hemopure and STS improved urine osmolality. Urine osmolality levels of renal grafts preserved for 24 h at 10°C in UW (*n* = 5), UW + 150 µM STS (*n* = 4), Hemopure + UW (*n* = 4), or Hemopure + UW + 150 µM STS (*n* = 5). Osmolality levels were measured using the Model 3320 Osmometer and normalized to company-provided standards. Bars represent mean ± SEM. Values were compared using one-way ANOVA followed by Tukey’s post hoc test. *, *p* < 0.05. **, *p* < 0.01. ****, *p* < 0.0001.

**Figure 7 ijms-25-02210-f007:**
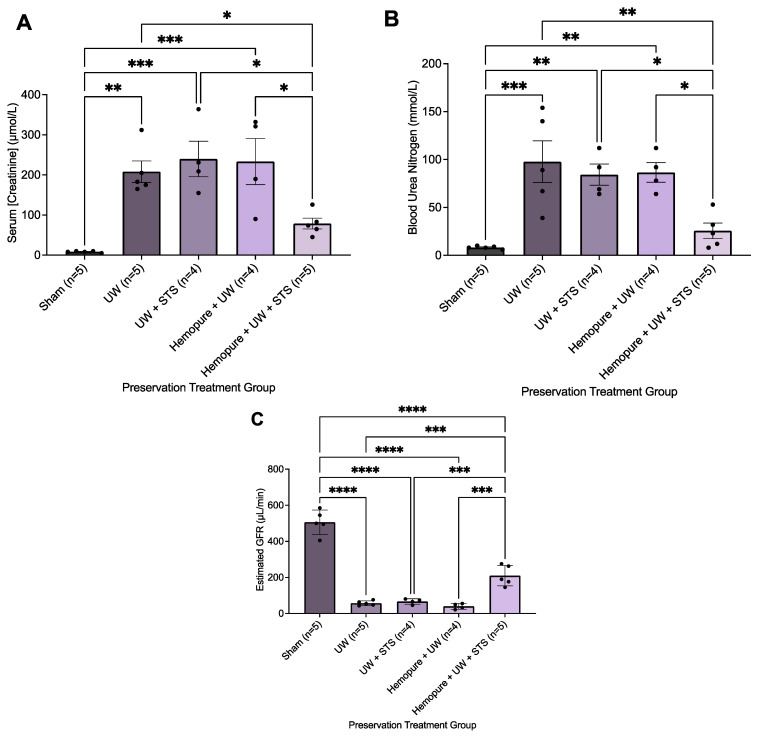
Renal graft preservation at 10 °C with Hemopure and STS enhanced recipient kidney function following transplantation. Serum creatinine (**A**), blood urea nitrogen (**B**), and estimated glomerular filtration rate (**C**) of kidney transplant recipients whose donor kidneys were perfused and stored for 24 h at 10 °C in UW (*n* = 5), UW + 150 µM STS (*n* = 4), Hemopure + UW (*n* = 4), or Hemopure + UW + 150 µM STS (*n* = 5), as well as those of Sham-operated rats. Values were analyzed via one-way ANOVA and Tukey’s post hoc test. *, *p* < 0.05. **, *p* < 0.01. ***, *p* < 0.001. ****, *p* < 0.0001.

**Figure 8 ijms-25-02210-f008:**
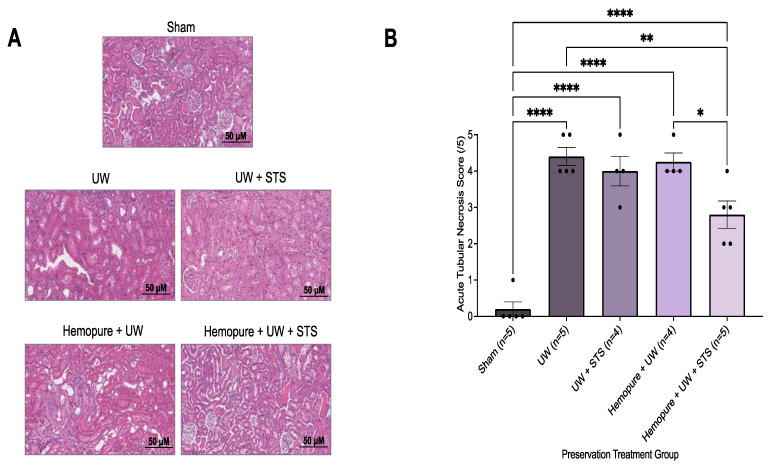
Renal graft preservation at 10 °C with Hemopure and STS reduced acute tubular necrosis after 24 h of storage. (**A**) Representative H&E images of formalin-fixed kidney sections on POD 3. Images were taken at 20× magnification (scale bar = 50 µM). (**B**) Acute tubular necrosis (ATN) scores assigned by two blinded renal pathologists (1 = <11%, 2 = 11–24%, 3 = 25–45%, 4 = 46–75%, 5 = >75%). Each individual data point indicates the score assigned to one rat kidney sample. Lines represent mean ± SEM. Values were compared using one-way ANOVA followed by Tukey’s post hoc test. *, *p* < 0.05. **, *p* < 0.01. ****, *p* < 0.0001.

**Figure 9 ijms-25-02210-f009:**
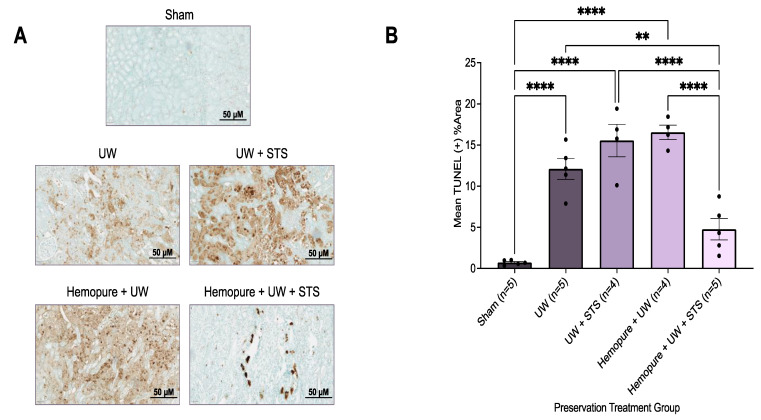
Renal graft preservation at 10 °C with Hemopure and STS reduced apoptotic tissue injury after 24 h of storage. (**A**) Representative TUNEL images of formalin-fixed kidney sections on POD 3. Images were taken at 20× magnification (scale bar = 50 µM). (**B**) Mean %TUNEL + area determined with ImageJ v.1.53 using a ratio of TUNEL + area (brown) to total tubular area. Each individual data point indicates the mean %TUNEL + area of 10 random fields of view of one rat kidney sample. Lines represent mean ± SEM. Values were compared using one-way ANOVA followed by Tukey’s post hoc test. **, *p* < 0.01. ****, *p* < 0.0001.

**Figure 10 ijms-25-02210-f010:**
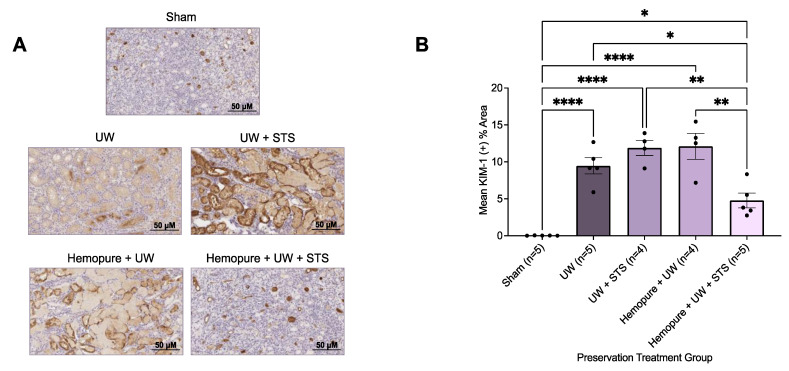
Renal graft preservation at 10 °C with Hemopure and STS mitigated kidney tissue injury after 24 h of storage. (**A**) Representative immunohistochemical images of formalin-fixed kidney sections on POD 3 stained with Kim-1, a biomarker of renal injury. Images were taken at 20× magnification (scale bar = 50 µM). (**B**) Mean %KIM-1+ area determined with ImageJ using a ratio of KIM-1+ area (brown) to total tubular area. Each individual data point indicates the mean %KIM-1+ area of 10 random fields of view of one rat kidney sample. Lines represent mean ± SEM. Values were compared using one-way ANOVA followed by Tukey’s post hoc test. *, *p* < 0.05. **, *p* < 0.01. ****, *p* < 0.0001.

**Figure 11 ijms-25-02210-f011:**
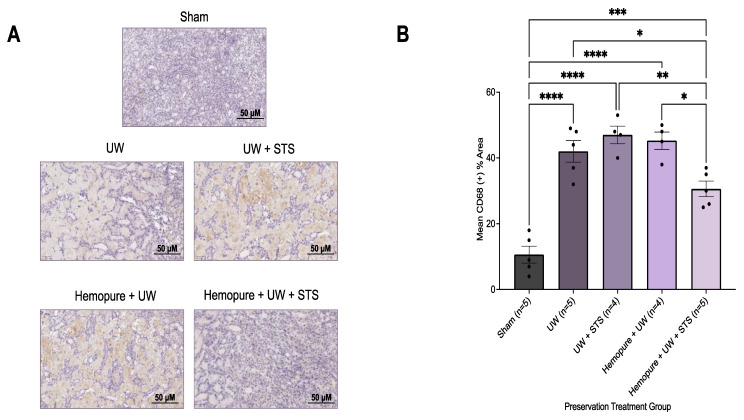
Renal graft preservation at 10 °C with Hemopure and STS reduced macrophage infiltration after 24 h of storage. (**A**) Representative immunohistochemical images of formalin-fixed kidney sections on POD 3 stained with CD68, a macrophage marker. Images were taken at 20× magnification (scale bar = 50 µM). (**B**) Mean %CD68+ area determined with ImageJ using a ratio of CD68+ area (brown) to total tubular area. Each individual data point indicates the mean %CD68+ area of 10 random fields of view of one rat kidney sample. Lines represent mean ± SEM. Values were compared using one-way ANOVA followed by Tukey’s post hoc test. *, *p* < 0.05. **, *p* < 0.01. ***, *p* < 0.001. ****, *p* < 0.0001.

**Figure 12 ijms-25-02210-f012:**
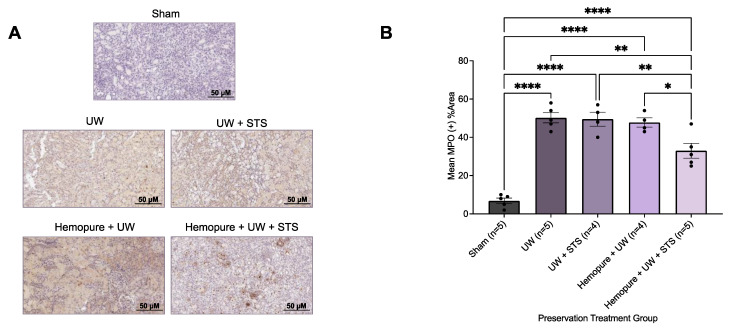
Renal graft preservation at 10 °C with Hemopure and STS decreased neutrophil infiltration after 24 h of storage. (**A**) Representative immunohistochemical images of formalin-fixed kidney sections on POD 3 stained with MPO, which is associated with neutrophil expression. Images were taken at 20× magnification (scale bar = 50 µM). (**B**) Mean %MPO+ area determined with ImageJ using a ratio of MPO+ area (brown) to total tubular area. Each individual data point indicates the mean %MPO+ area of 10 random fields of view of one rat kidney sample. Lines represent mean ± SEM. Values were compared using one-way ANOVA followed by Tukey’s post hoc test. *, *p* < 0.05. **, *p* < 0.01. ****, *p* < 0.0001.

## Data Availability

The original contributions presented in the study are included in the article.

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
