# Peer review of "Evaluating the Effects of Kidney Preservation at 10 °C with Hemopure and Sodium Thiosulfate in a Rat Model of Syngeneic Orthotopic Kidney Transplantation"

_ijms, 2024, doi:10.3390/ijms25042210_

Round 1

Reviewer 1 Report

Comments and Suggestions for Authors

The authors sought to investigate whether SCS at 10ËšC with sodium thiosulfate (STS) and Hemopure, a blood substitute, provided similar graft protection to STS at 4ËšC.

The main problem with the current manuscript is that a number of superfluous results are presented, but some basic results needed for making meaningful comparisons are missing. The main question for the reader is how the protective effects of STS + Hemopure compare to the large number of previously published interventions.

In their previous article (doi: 10.1016/j.biopha.2023.115549) the authors showed that approximately 90 % of rats transplanted with STS-treated kidneys at 4 C survived up to 14 days after transplantation compared with poor survival of rats transplanted with UW-preserved kidneys, i.e. STS provided an excellent graft protection at 4 C. However, the current results show that all rats transplanted with STS-treated kidneys at 10 C died by day 8 after transplantation. These results clearly show that preservation at 4 C is definitely and indisputably better than at 10 C, when the kidney is treated with STS. The main results of the previous study are carefully hidden in the current manuscript, which - in my opinion - deserves immediate rejection.

The basic observation was that “There was no significant difference between the different renal preservation treatments in recipient survival following kidney transplantation”. In other words, the interventions the authors used were INEFFECTIVE. The groups transplanted with UW + STS + Hemopure- and UW-preserved kidney had the same survival. It means the treatment with STS + Hemopure at 10 C is not any better than UW alone.

Moreover, rats transplanted with UW + Hemopure- and UW + STS-preserved kidneys died in 4 and 8 days. How on earth can the additive effect of two negative treatments be positive?

What is the explanation for the observation that rats transplanted with UW + STS + Hemopure-preserved had similar survival to those transplanted with UW-preserved kidneys if the former treatment had a clear protective effect. Why this protective effect is short lasting?

The introduction focuses on a number of issues not closely related to the topic of the article. The authors should delete most parts of the general statements on the need and superiority of kidney transplantation and demonstrate the various preclinical attempts to improve kidney preservation.

As all cells were treated with STS, no conclusions can be drawn about the effects of STS as there are no untreated cells to compare the results with. It is logically possible, although unlikely, that STS reduced cell viability.

The temperatures chosen for cell culture experiments should be better explained. Why 37 C was included, which is obviously not suitable for long-term preservation. I miss 7 and 13 C, temperatures above 13 C are not reasonable.

Comments on the Quality of English Language

No relevant.

Reviewer 2 Report

Comments and Suggestions for Authors

Dear Colleagues,

it was a great pleasure to read your paper. As a practicing surgeon who deals with kidney transplants on a daily basis, I read it with great interest. From an editorial point of view, it seems more beneficial, to change the order of the "methods" and "discussion" sections. In the document provided to me, the discussion comes before the methods section, which is an unusual solution and makes review difficult. In the conclusions section, the following sentence can be found: This novel circuit could shift the paradigm away from the costly and North American-exclusive setup of machine perfusion to a device that is more cost-effective and widely accessible for nations around the globe. In my department, we use machine perfusion on a daily basis, and we are not North Americans. Correct me if I'm wrong, are you suggesting that your solution can replace pulsed perfusion? Machine perfusion has an advantage over simple hypothermia, which results from, among other things - the effect on the vascular endothelium, etc. Do you think that your modified fluids can be used in perfusion pumps, with the assumption that the organ will be stored at a higher temperature? This should be clarified.

I wish you all the best!

Kind Regrads, 

Round 2

Reviewer 1 Report

Comments and Suggestions for Authors

Thank you for considering my recommendations.

I am still convinced that the beneficial effects emphasized in the manuscript are worthless if survival is not prolonged. You did not try to explain why much better initial kidney function does not prolong survival. I can see that you are motivated in emphasizing the positive results and not interested in correct reporting and conclusion. You still have the option to change this approach if you want.

It is written in the discussion that “eGFR estimates the flow rate”, eGFR estimates filtration rate, please correct.

The figures have been renumbered, but you have not corrected the reference numbers for the figures.
